# Numerical Comparison of Restored Vertebral Body Height after Incomplete Burst Fracture of the Lumbar Spine

**DOI:** 10.3390/jpm12020253

**Published:** 2022-02-10

**Authors:** Guan-Heng Jhong, Yu-Hsuan Chung, Chun-Ting Li, Yen-Nien Chen, Chih-Wei Chang, Chih-Han Chang

**Affiliations:** 1Department of Biomedical Engineering, National Cheng Kung University, Tainan 701, Taiwan; jguanheng@gmail.com (G.-H.J.); changbmencku@gmail.com (C.-H.C.); 2Department of Orthopedics, Show Chwan Memorial Hospital, Changhua 500, Taiwan; supersam9101005@gmail.com; 3Institute of Geriatric Welfare Technology & Science, Mackay Medical College, New Taipei 252, Taiwan; ctli0412@mmc.edu.tw; 4Department of Physical Therapy, Asia University, Taichung 413, Taiwan; 5Department of Orthopedics, National Cheng Kung University Hospital, College of Medicine, National Cheng Kung University, Tainan 704, Taiwan; 6Department of Orthopedics, College of Medicine, National Cheng Kung University, Tainan 701, Taiwan

**Keywords:** biomechanics, burst fracture, kyphoplasty, stress, SpineJack, vertebral body height

## Abstract

Background and objectives: Vertebral compression fracture is a major health care problem worldwide due to its direct and indirect negative influence on health-related quality of life and increased health care costs. Although a percutaneous surgical intervention with balloon kyphoplasty or metal expansion, the SpineJack, along with bone cement augmentation has been shown to efficiently restore and fix the lost vertebral height, 21–30% vertebral body height loss has been reported in the literature. Furthermore, the effect of the augmentation approaches and the loss of body height on the biomechanical responses in physiological activities remains unclear. Hence, this study aimed to compare the mechanical behavior of the fractured lumbar spine with different restored body heights, augmentation approaches, and posterior fixation after kyphoplasty using the finite element method. Furthermore, different augmentation approaches with bone cement and bone cement along with the SpineJack were also considered in the simulation. Materials and Methods: A numerical lumbar model with an incomplete burst fracture at L3 was used in this study. Two different degrees of restored body height, namely complete and incomplete restorations, after kyphoplasty were investigated. Furthermore, two different augmentation approaches of the fractured vertebral body with bone cement and SpineJack along with bone cement were considered. A posterior instrument (PI) was also used in this study. Physiological loadings with 400 N + 10 Nm in four directions, namely flexion, extension, lateral bending, and axial rotation, were applied to the lumbar spine with different augmentation approaches for comparison. Results: The results indicated that both the bone cement and bone cement along with the SpineJack could support the fractured vertebral body to react similarly with an intact lumbar spine under identical loadings. When the fractured body height was incompletely restored, the peak stress in the L2–L3 disk above the fractured vertebral body increased by 154% (from 0.93 to 2.37 MPa) and 116% (from 0.18 to 0.39 MPa), respectively, in the annular ground substance and nucleus when compared with the intact one. The use of the PI could reduce the range of motion and facet joint force at the implanted levels but increase the facet joint force at the upper level of the PI. Conclusions: In the present study, complete restoration of the body height, as possible in kyphoplasty, is suggested for the management of lumbar vertebral fractures.

## 1. Introduction

Vertebral compression fracture due to osteoporosis affects 1.4 million patients with osteoporosis per year worldwide [1]. In patients with fragility fractures, 35% had prevalent vertebral compression fractures [2]. Osteoporosis is a systemic disorder that decreases the strength and elastic modulus of the bone and increases the risk of bone fractures [3]. Vertebral compression fracture is a major health care problem worldwide due to its direct and indirect negative influence on health-related quality of life and increased health care costs [4,5,6]. Traditional treatments for vertebral compression fractures include bed rest, medicine, braces, and physical therapy. However, one-third of patients have been reported to have progressive functional limitations. Furthermore, immobilization due to bed rest aggravates osteoporosis and may predispose the patient to future fractures [7].

A minimally invasive surgical intervention to fix the fractured body has been reported to achieve immediate pain relief and recovery of daily activity [8,9]. Percutaneous vertebroplasty is a surgical approach for the fixation of the fractured vertebral body via percutaneous injection of bone cement (most commonly polymethyl methacrylate (PMMA)) to improve clinical outcomes [10]. However, the structural deformity, including the loss of body height of the fractured body and the spinal curve, is not restored after vertebroplasty; furthermore, the bony fragment might bulge into the spinal canal in unstable fractures [11].

To correct the deformed spinal curve after vertebral fracture, other percutaneous techniques, such as balloon kyphoplasty (BKP) and metal expansion (the SpineJack), have been developed. In BKP, a balloon is used to expand the collapsed vertebral body, and the restored space is filled with bone cement for fixation and augmentation. The SpineJack, which contains a central screw and two deployable plates, is an intravertebral expandable device. Bone cement is also filled into the space of the fractured body created by the SpineJack. To date, the SpineJack has been proven to relieve pain efficiently in the management of acute vertebral compression fractures.

Although the BKP and SpineJack have been shown to restore the lost vertebral body height, 21%–30% vertebral body height loss has been reported in the literature [12,13]. The biomechanical behavior of the lumbar spine is highly related to its geometry, from a biomechanical perspective. Hence, the biomechanical behaviors of the lumbar spine without complete restoration are different from those with complete restoration and even an intact lumbar spine. However, to date, changes in the biomechanical behaviors of the fractured lumbar spine without complete restoration of the body height are not completely clear. Furthermore, loss of vertebral height is associated with changes in the biomechanical properties of the spine, and this is incompletely understood.

Hence, this study aimed to compare the mechanical responses of the lumbar spine with L3 vertebral fractures and different restored body heights. Furthermore, different augmentation approaches with bone cement, bone cement along with the SpineJack, and a posterior instrument (PI) for the fractured lumbar spine were also considered in this study.

## 2. Materials and Methods

An intact lumbar FE model containing L1–L5 was used for the study simulation. First, the model was validated by comparing the results of the present intact lumbar FE model with those of the published FE and cadaveric models under identical loading conditions. After validation, the intact model was modified with an incomplete burst fracture at the L3 vertebral body and then augmented with bone cement, SpineJack and bone cement, and PI.

### 2.1. Solid Model

A 3D intact lumbar L1–L5 model (Figure 1) was developed based on the computed tomography (CT) images of a 30-year-old healthy man without osteoporosis, with a body weight of 70 kg and body height of 170 cm. The solid model of the cortical and cancellous bones of each vertebral body were rendered by the bony contours, which were retrieved according to the different gray values of the cortical and cancellous bones in each CT image. The intervertebral disks were created as the spaces between the vertebral bodies.

This study investigated a worst-case condition of incomplete burst fracture of the lumbar spine with partial bone loss after fracture due to severe osteoporosis. Hence, the lumbar L3 was entirely cut into two separate parts with a virtual plane (Figure 2). Then the partial volume of the anterior cortex of the L3 was removed to represent its discontinuity after kyphoplasty and augmentation. The anterior portion of the cancellous bone with a 5-mL volume was identified as the bone cement [14]. In addition to the bone cement, a titanium SpineJack (Stryker, MI, USA) and PI (pedicel screw and rod) were also used along with the bone cement (Figure 2). The SpineJack, with full vertical expansion, was placed bilaterally inside the vertebral body of the L3. Because the vertebral body was separated into two parts, a posterior instrument, including two pedicle screws and rods, was used to achieve stable fixation of the lumbar spine. The pedicle screws were inserted by passing through the bilateral pedicles of L2 and L4. Two metallic rods were used to connect the upper and lower pedicle screws on the same side (Figure 2).

The expansion height, plate length, insertion diameter, and blocking tube diameter were 17, 19, 5, and 2.5 mm, respectively. The outer diameter of the pedicle screws and the rod were both set to 6.5 mm, and the length of the rod was set to 50 mm. The geometry of the pedicle screw was defined according to the commercial product. In addition to the models without body height loss, incomplete restoration of the body height of the lumbar spine was also considered. A wedge-shaped bone loss of 20% at the anterior portion of the vertebral height has been developed [2,12]. The anterior cortex was removed in the same manner as in the model with complete body restoration.

In total, an intact lumbar spine and six different fractured and augmentation lumbar models were developed (Figure 1), including complete restoration of the vertebral body augmented with bone cement (CC), bone cement along with PI (CCPI), bone cement along with the SpineJack (CCSJ), bone cement along with the SpineJack and PI (CCSJPI), incomplete restoration of the vertebral body augmented with bone cement (ICC), and bone cement along with PI (ICCPI).

### 2.2. Finite Element Model

The solid model was imported into ANSYS Workbench 2019 R3 for mesh generation and further simulations. Quadratic elements (solid 187) were used to mesh the entire lumbar model. The length of the element edge was set to 3 mm globally. The mesh density of the intervertebral disk was locally refined by reducing the length of the element edge to 1 mm with the command “sizing” in ANSYS Workbench. The length of the element edge for the SpineJack and pedicel screw and the bone area surrounding the implants were set to 1 mm. The facet joints were set to a frictional surface to surface contact with a coefficient of 0.1 [15]. The contact behaviors between the implants (SpineJack and pedicel screw), bone, and bone cement as well as between the pedicle screw and the metallic rod were set to bond.

The spinal ligaments, namely, the anterior longitudinal ligament, posterior longitudinal ligament, supraspinous ligament, interspinous ligament, intertransverse ligament, ligamentum flavum, and joint capsule, were represented by tension-only springs. The locations of the ligaments were defined according to their origin and insertion sites. The stiffness of the springs (Table 1) as the ligaments were set according to the literature [16,17,18]. The material properties (Table 2) of the bone were set as osteoporosis, and the parameters were defined according to the literature [18,19].

### 2.3. Validation and Boundary Condition

To confirm the reliability of the present FE model, a 10 Nm pure moment in four different directions (flexion, extension, lateral bending, and axial rotation) was applied to the intact lumbar FE model. The total ROMs around the major principal axis were compared with those in the published FE models and cadaveric tests for validation [20,21,22]. Furthermore, the intradiscal pressure of the disks and the force on the facet joints were compared with those in the published FE models under identical loading conditions [23]. Additionally, the results of normalized ROMs of the CC and CCPI were compared with those in the literature [24] to validate the present model with kyphoplasty. The mesh density of the CCSJPI was also globally increased for the convergence test. The number of nodes increased from 2,676,568 to 14,333,320, and the peak equivalent stress (also called von Mises stress) of the SpineJack and PI in 400 N axial compression were used as indices for the convergence test. After the validation and convergence test, the physiological load of a 400 N vertical force was applied to the superior surface of L1 in the first step [25], and then a 10 Nm pure moment in four different directions, namely flexion, extension, lateral bending, and axial rotation (Figure 3), was applied in the second step [26].

### 2.4. Index

To compare the mechanical responses of the fractured vertebral body with different restored body heights and augmented with bone cement, SpineJack, and PI, the results of the total ROMs (L1–L5) and the ROM at each level in the four directions, the contact force on the facet joint, and the intradiscal stress of the intervertebral disks were plotted for comparison. The equivalent stresses of the metallic implants were also compared.

## 3. Results

### 3.1. Results of Validation

The results of the ROMs of the present FE model at 10 Nm pure moment on the three principal planes were similar to those in previous studies [20,21,22,24] (Figure 4). Furthermore, the median intradiscal pressure and facet joint force in the present intact FE model were similar to those in previous studies (Figure 4) [23]. The disk pressure and facet joint force were the highest in flexion and in axial rotation, respectively. The normalized ROMs after kyphoplasty with bone cement and bone cement with PI were similar to those reported in the literature [24]. The difference in the peak stress of the SpineJack and PI with different mesh densities were 4.6% and 0.7%, respectively (Figure 5).

### 3.2. Range of Motions

The total ROM of the ICC increased in extension but decreased in flexion compared with that of the intact lumbar region (Figure 6), whereas the summation of the total ROM in flexion and extension was similar in the ICC (20.8°) and intact (19°) lumbar regions. Furthermore, the total ROMs of the CC and CCSJ were similar to those of the intact lumbar spine. The use of PI with bone cement or bone cement together with the SpineJack was expected to reduce the ROM at the levels with the PI but increase the ROM below the PI. The ROMs of disk 4 in lateral bending increased by 83%, 82%, and 53% in the CCPI, CCSJPI, and ICCPI, respectively, when compared with that of the intact lumbar.

### 3.3. Force on the Facet Joint

The contact force on the L23 facet joint of the ICC was obviously increased in extension, lateral bending, and axial rotation when compared with that of the intact lumbar spine (Figure 5). The force on the L23 facet joint in the ICC increased by 47% (from 122.8 to 180.5 N) in extension when compared with the intact lumbar spine. Using the PI could bypass the loading through the PI instead of through the facet joints. Hence, the force on the facet joint with the PI was almost zero. In contrast, the force on the lower level, the L45 facet joint, in lateral bending increased by 18.4%, 17.5%, and 16.5% in the CCPI, CCSJPI, and ICCPI, respectively compared with that of the intact lumbar spine.

### 3.4. Equivalent Stress of the Disk

The equivalent stress of disk 2, both the annular ground substance and the nucleus, in the ICC was much higher than that of the intact lumbar spine in extension and lateral bending (Figure 7, Figure 8 and Figure 9). The peak stress of disk 2 in the ICC increased by 154% (from 0.93 to 2.37 MPa) and 116% (from 0.18 to 0.39 MPa), respectively, in the annular ground substance and nucleus when compared with the intact one. Using the PI could protect the disk and drastically reduce the disk stress. The peak stress of the disk 2 nucleus in extension was reduced by 77%, 77.2%, and 78.9% in the CCPI, CCSJPI, and ICPI, respectively, when compared with that in the intact lumbar region.

### 3.5. Equivalent Stress of the Implant

The highest equivalent stress of the SpineJack occurred in lateral bending without the PI, but in flexion with the PI (Figure 10). The peak stress of the SpineJack decreased by 46.5% (from 71 to 38 MPa) after using the PI. The highest equivalent stress of the PI was revealed in axial rotation without the SpineJack (ICPI), whereas the lowest stress occurred in extension with incomplete body height restoration (ICC). The peak stress of the PI in the ICC were 317.7, 121.1, 266.4, and 560.8 MPa in flexion, extension, lateral bending, and axial rotation, respectively (Figure 11).

## 4. Discussion

Vertebral fracture is a common disorder in elderly people, particularly in those with osteoporosis. The disorder not only affects the quality of life, but also threatens their life. The pain caused by the fracture reduces daily activity, mobility, and cardiopulmonary function. Increasing cardiopulmonary function with exercise improves the quality of life [27]. Hence, during fracture management, it is important to allow patients to return to their original daily activities as early as possible. In the present study, augmentation with both bone cement and bone cement along with the SpineJack could efficiently restore mechanical behaviors similar to those of an intact lumbar spine. Furthermore, more stability was achieved when PI was used along with augmentation. The results suggest the need for surgical intervention and augmentation after vertebral fractures.

In the present study, a simplified vertebral fracture model was developed to compare the responses of the collapsed vertebral body with and without full restoration. Because the geometry of the bone cement was simplified as the shape of the cancellous bone, the stiffness of the restored body in the present model was higher than the real one. The stiffness was higher because the elastic modulus of the bone cement was higher than that of the cancellous bone. Additionally, as more bone cement was used in the present model, the stiffness was higher. The disk is the major deformed part of the spine under loading, hence the effect of higher stiffness of the body on the lumbar spine is minor. To confirm the effect of the simplified shape of the bone cement, the normalized ROMs of the implanted levels were compared to the published mode and the maximum difference was 11%.

Vertebral fractures are highly related to osteoporosis, and the problems associated with vertebral fractures are serious socioeconomic problems, including the increased cost of medication and health care [28]. In addition, a comparison between the cost due to loss of ambulation and medication for osteoporosis was statistically significant [29]. Hence, reducing the pain promptly after the fracture and allowing the patients back to daily activities are important in the management of vertebral fractures. Vertebral augmentation with bone cement augmentation to recapture mobility has been reported to reduce mortality compared to pain palliation [30]. The mortality rate of vertebral compression fracture with surgical intervention (vertebroplasty and/or kyphoplasty) 10 years after the fracture was reported to be 22% lower than that without intervention [31]. In another study, vertebral augmentation with surgical intervention was reported to result in lower morbidity and mortality than nonsurgical interventions [32]. Early fixation of vertebral fractures was suggested 72 h after the onset of the fracture [33].

Incomplete restoration of the fractured vertebral body, particularly with BKP, is a possible result in the management of vertebral fractures. In the present simulation, although the difference in total ROM in flexion-extension between the intact lumbar spine and ICC was just 1.8°, the change in the pattern of ROM should not be ignored. The ROM of the ICC decreased during flexion but increased during extension. Furthermore, the stress of the nucleus and annular ground substance in the ICC was 2 and 2.3 times higher than that in the intact lumbar spine, respectively. When the collapsed vertebral body was not fully restored after the fracture, the shape of the fractured lumbar spine was different from that of the intact lumbar spine. The curve was reduced, and the spine became straighter than the intact lumbar spine. The deformed structure also triggered abnormal mechanical responses of the lumbar spine. In the intact lumbar spine, the deformation is distributed in every disk under loading, whereas in a deformed and straight lumbar spine, the deformation is more concentrated in a specific area. In the present simulation, the changed force distribution increased the loading on the facet joint and disk stress of disk 2. A dramatically increased disk stress is a risk factor for disk generation. In the literature, disk degeneration at the adjacent level of the fractured vertebral body was observed in MR images [34]. In addition, percutaneous kyphoplasty and augmentation can significantly reduce disk generation at the adjacent level, compared with that without percutaneous kyphoplasty and body height restoration [35].

Kyphoplasty with balloon and metal expansion (SpineJack) were both used to restore vertebral height loss for vertebral fractures. In the present simulation, the differences in the ROMs, disk stress, facet joint force, and implant stress between bone cement and bone cement along with the SpineJack were not obvious. In the presentation, we just compared the structural stiffness of bone cement and bone cement with a metal framework with identical geometry, and the difference in expansion was not considered. Hence, the results were the final status of the lumbar spine with body height restoration, and the difference in the process of expansion with the balloon and SpineJack was not considered. The metal framework was left inside the body and augmented with bone cement, without the balloon. In the present simulation, the effect of augmentation for the body with bone cement was similar to that with bone cement along with the metal framework. Although the elastic modulus of the used metal titanium in the simulation was much higher than that of bone cement, the volume of the metal framework was much lower than that of the bone cement. Hence, the difference between the augmentation with and without the metal framework is very minor.

In previous studies, the SpineJack has been reported to restore the loss of body height compared to that of balloon dilation. The SpineJack could preserve the maximum height gain (mean 1% height loss) better than BKP (mean 16% height loss) [14]. In addition, the cement volume used with the SpineJack was lower than that with kyphoplasty [14]. In another study, the SpineJack was reported to restore the vertebral body height (1.14 ± 2.61 mm) immediately after the surgery compared to BKP (0.31 ± 2.22 mm) [36]. Furthermore, even 12 months after the surgery, the restored body height with the SpineJack (1.31 ± 2.58 mm) was still higher than that with BKP (0.10 ± 2.34 mm) [36]. Moreover, in a study of safety and clinical performance, the SpineJack was demonstrated to maintain the restored body height and lumbar curve compared to that of BKP [37,38]. Although the SpineJack demonstrated better clinical results than BKP, the cost of the SpineJack is much higher than that of BKP. The high cost makes it too costly for economically disadvantaged persons.

The use of PI could share the loading of the fractured vertebral body and then reduce the stress of the nuclear and annular ground substances. However, the ROMs and disk stress at the lower adjacent level in the simulation increased. These results are in accordance with published research [39,40]. Although the decrease in the ROM and the increase in the disk stress were not very obvious in each loading, the incidence of the degeneration at the adjacent level of the PI has been reported in the literature [41,42,43,44]. Surgeons and physical therapists must consider the risk of adjacent level degeneration and take proper precautions, such as patient education, muscle strengthening, and even removal of the PI. Although the use of PI increased the ROMs and disk stress at the adjacent level, the PI protected the lumbar spine without complete restoration of body height and then reduced the ROMs and disk stress. Hence, PI is a suggested option for fractured lumbar spine without complete restoration of body height.

This study has some limitations. First, the present model was built from a young, healthy adult. The difference in geometry between the degenerated and healthy lumbar spine was not considered. Second, the force on the lumbar developed by the fracture occurrence and the balloon and SpineJack expansion were not considered. In the present simulation, only the final status of the lumbar spine with kyphoplasty and various augmentation approaches were considered. Third, a worst-case condition of osteoporosis was considered; hence, the anterior part of the cancellous bone was entirely substituted with bone cement. The geometry of the bone cement was different from that of the actual structure. Fourth, the ligaments and annulus fibrosus were simplified as one-dimensional spring elements. The deformation of the cross-sectional area is not represented in the present simulation. Furthermore, the soft tissues were healthy vertebrae instead of osteoporotic vertebrae. Finally, the present simulation is a simplified and idealized mathematical simulation. The gap between idealization and reality must still be considered in clinical applications.

## 5. Conclusions

In this study, an FE model of the lumbar spine with an incomplete burst fracture augmented with bone cement and a metal framework (SpineJack) along with bone cement and PI was developed. The present model demonstrated the biomechanical responses of the collapsed vertebral body with complete and incomplete restorations of the body height and augmentation with bone cement, a SpineJack, and PI. Our results revealed that complete restoration of the vertebral body height with kyphoplasty and augmentation with bone cement and SpineJack is recommended for the lumbar spine with an incomplete burst fracture to recapture the biomechanical behavior of the intact lumbar spine. Incomplete restoration of the collapsed body height should be considered inferior to complete restoration to avoid an increase in disk stress. Furthermore, PI is also suggested for the lumbar spine with incomplete restoration.

## Figures and Tables

**Figure 1 jpm-12-00253-f001:**
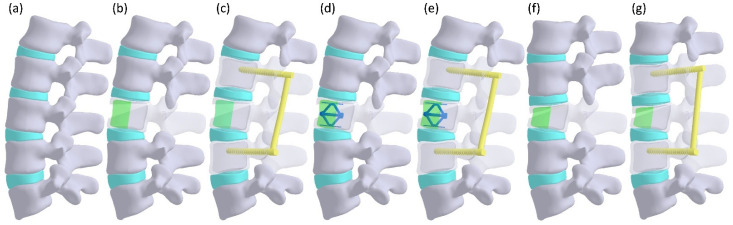
The models used in this study. (**a**) Intact lumbar spine, (**b**) L3 body height completely restored with bone cement (CC), (**c**) bone cement along with PI (CCPI), (**d**) bone cement along with SpineJack (CCSJ), (**e**) bone cement along with SpineJack and PI (CCSJPI), (**f**) L3 incompletely restored with bone cement (ICC), and (**g**) bone cement along with PI (ICCPI).

**Figure 2 jpm-12-00253-f002:**
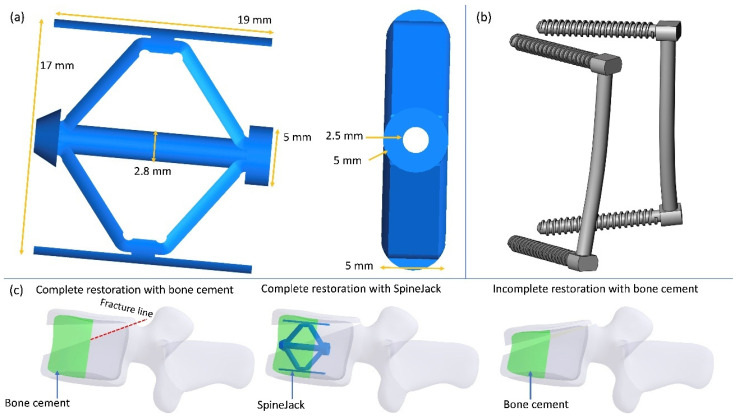
The (**a**) SpineJack, (**b**) PI, and (**c**) fractured vertebral body used in this study.

**Figure 3 jpm-12-00253-f003:**
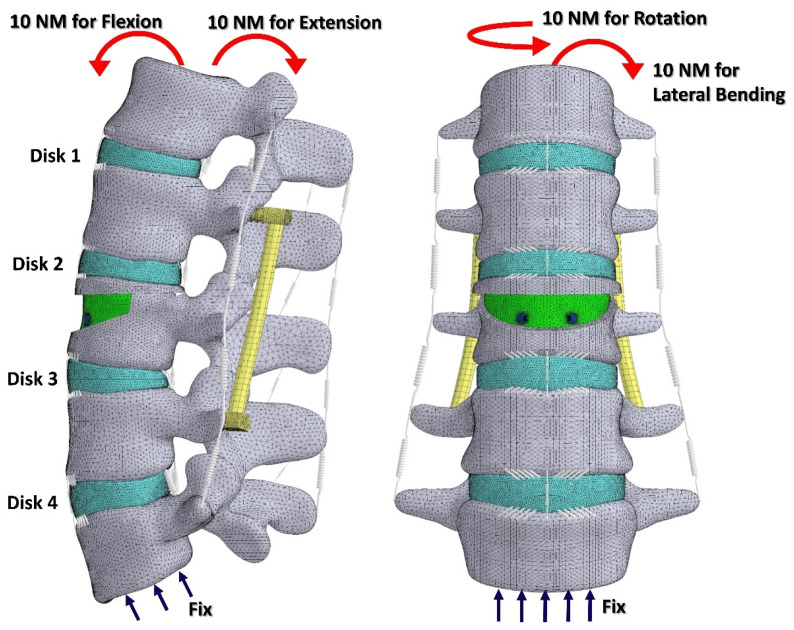
The finite element model and boundary conditions in this study.

**Figure 4 jpm-12-00253-f004:**
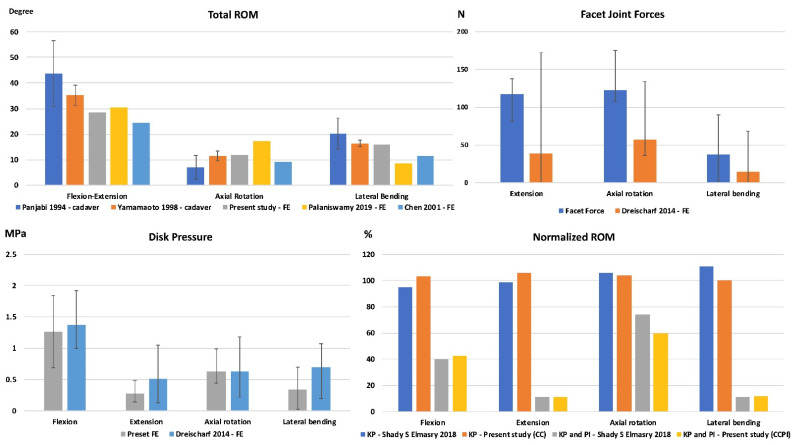
The results of validation of total ROMs, facet joint force and peak disk stress (median, maximum, and minimum of the four disks), and normalized ROM with bone cement and SpineJack.

**Figure 5 jpm-12-00253-f005:**
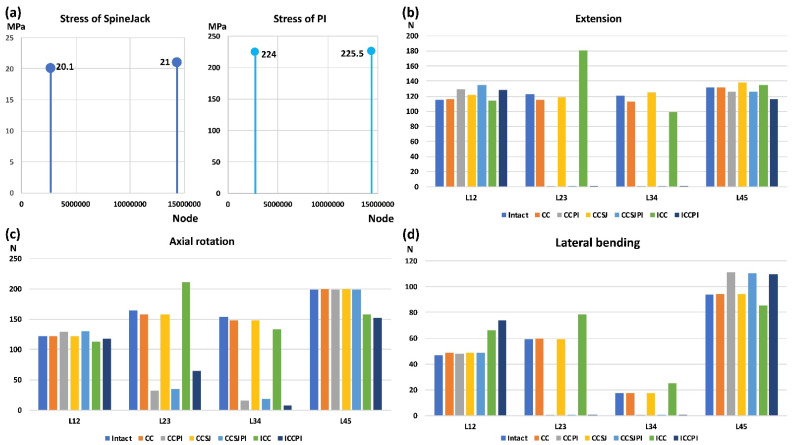
The results of the convergence test (**a**) and force on the facet joint in extension (**b**), axial rotation (**c**) and lateral bending (**d**).

**Figure 6 jpm-12-00253-f006:**
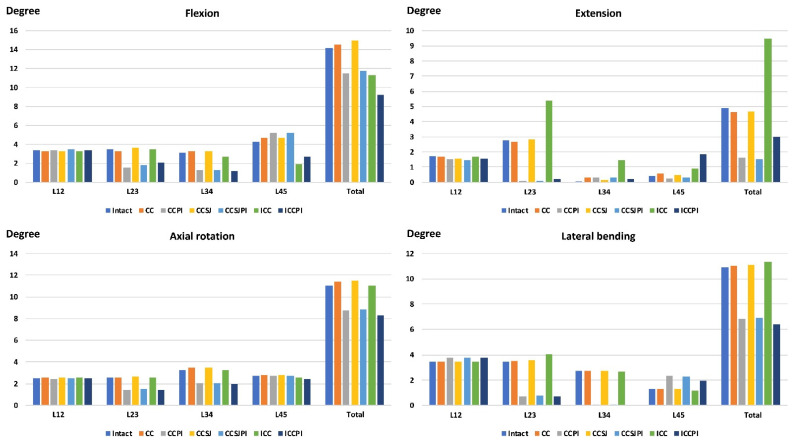
The results of ROMs.

**Figure 7 jpm-12-00253-f007:**
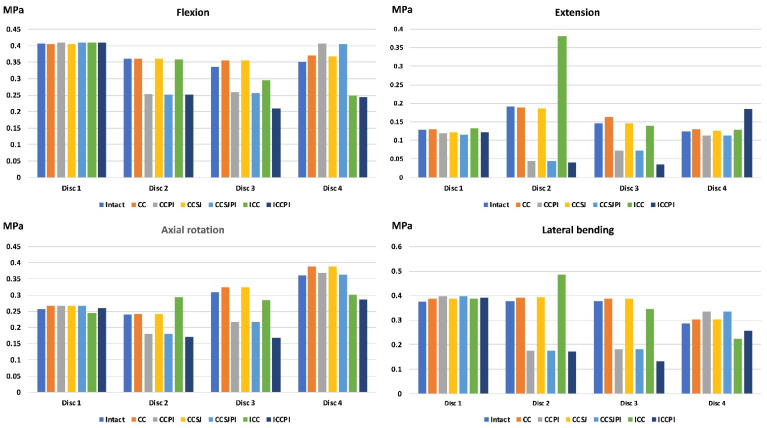
The results of nucleus stress.

**Figure 8 jpm-12-00253-f008:**
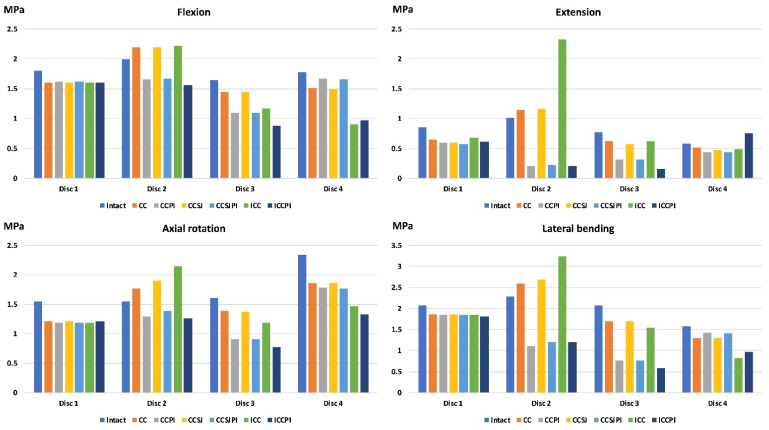
The results of the annular ground substance stress.

**Figure 9 jpm-12-00253-f009:**
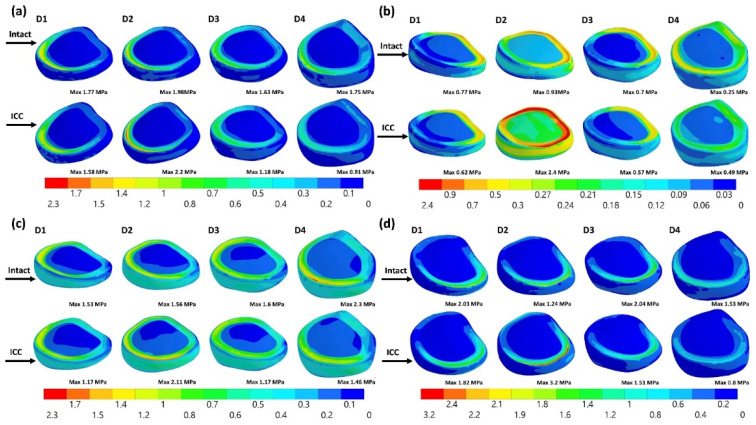
The equivalent stress of the disks in the ICC and intact lumber in flexion (**a**), extension (**b**), axial rotation (**c**), and lateral bending (**d**).

**Figure 10 jpm-12-00253-f010:**
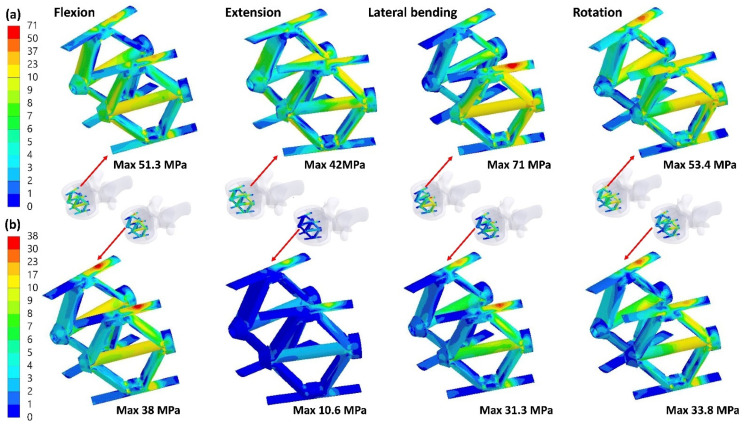
The equivalent stress of the SpineJack without (**a**) and with (**b**) the PI.

**Figure 11 jpm-12-00253-f011:**
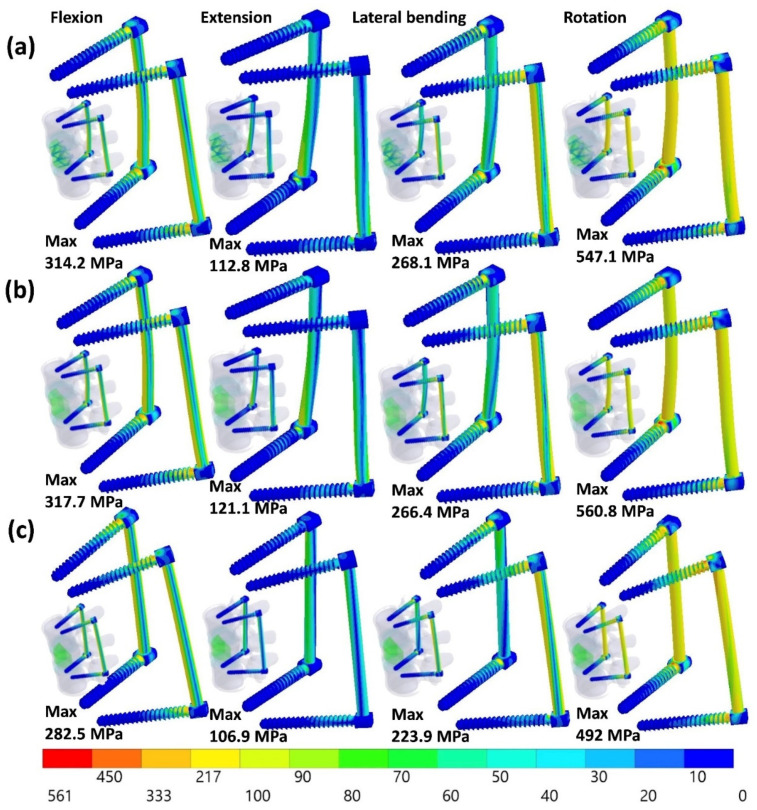
The equivalent stress of the PI in CCSJPI (**a**), CCPI (**b**), and ICPI (**c**).

**Table 1 jpm-12-00253-t001:** The stiffness of the ligaments and annular fiber used in the simulation.

Ligament	Stiffness (N/mm)	Spring Numbers at Each Level
Anterior longitudinal	210	1
Posterior longitudinal	20.4	1
Joint capsule	33.9	6
Ligament flavum	27.2	2
Interspinous ligament	11.5	1
Supraspinous ligament	23.7	1
Intertransverse ligament	50	2
Annular fiber	14	10

**Table 2 jpm-12-00253-t002:** The material properties used in the simulation.

Material	Elastic Modulus (MPa)	Poisson’s Ratio
Cortical bone	8040	0.3
Cancellous bone	34	0.3
Posterior element	2345	0.3
Nucleus pulposus	1	0.49
Ground substance	3.5	0.45
Cement	2600	0.3
Titanium (posterior instrument and SpineJack)	110 000	0.3

## Data Availability

The data used to support the findings of this study are included within the article.

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
