# Peer review of "Numerical Comparison of Restored Vertebral Body Height after Incomplete Burst Fracture of the Lumbar Spine"

_jpm, 2022, doi:10.3390/jpm12020253_

Round 1
Reviewer 1 Report
Summary
A finite element study is presented investigating the effect of kyphoplasty, SpineJack, and posterior instrumentation on the range of motion, facet joint contact forces, and intervertebral disc stress in the lumbar spine after osteoporotic vertebral body fracture. The results of this study are quite interesting, however not substantiated by a sufficient validation of the used model and therefore not of clinical value. Overall, the chosen study design is not suitable for this research question, while an experimental study design is required to fully replicate quasi-realistic conditions, such as the fracture morphology or the kyphoplasty. In total, there are too many rough assumptions in the methodology such as the fracture morphology, bone cement distribution and location, restoration height, mesh size, etc. A finite element model would just be useful in this case for expanding specific experimental results regarding facet joint contact forces or stresses in the intervertebral discs, which nevertheless requires a fully validated model for all experimentally determined conditions. Moreover, simulating solely one person-specific spine might not be sufficient for deriving clinically relevant conclusions, although the reviewer is aware of the effort underlying the creation of a finite element model.
Abstract
Line 24: Percent numbers should be rounded to so that no decimal positions are displayed.
Line 35: The unit ‘Newton meter’ should be abbreviated as Nm instead of NM (see also lines 153, 158, and 172).
Line 36: When describing spinal motions, the term ‘axial rotation’ should be used instead of ‘rotation’.
Introduction
Line 77: See comment for line 24.
Line 87: ‘fractured lumbar [spine]’ (see also lines 102, 125, 185, 192, 196, 199, 250, 254, and 296).
Methods
Line 96: Was this a healthy young or an old man with osteoporosis? There should be given more information on this person.
Lines 102-104: Were bone cement distribution and incomplete restoration simulated arbitrarily?
Lines 105-106: In osteoporotic vertebral bodies there is usually no burst fracture. Also, in this study, there was no burst fracture simulated, but some kind of open wedge fracture. Was the simulation of the fracture performed according to a classification system for osteoporotic vertebrae or just arbitrarily?
106-108: In real-life kyphoplasty, there is usually not such a large gap between the anterior bone fragments. Otherwise, the bone cement would completely run into the stomach, leading to severe complications in the patient!
108-109: Bone cement distribution does also not look very realistic as seen on Figure 2. Usually, the kyphoplasty produces either two ovoid bone cement clusters symmetrically to the sagittal plane (bipedicular balloon kyphoplasty) or one big central ovoid bone cement cluster (monopedicular radiofrequency kyphoplasty).
Lines 112-115: Were the pedicle screws and rods self-made or simulated from existing ones?
Lines 118-120: Were these lengths chosen arbitrarily or did the authors ask a surgeon?
Lines 134-138: How were the element sizes determined? Was there any senstitivity analysis?
Lines 145-147: The presented ligament stiffness values were derived from healthy people and do not reflect elderly (osteoporotic) vertebrae.
Lines 153-160: Was the validation of the model solely performed for the intact model without preload, fracture, bone cement, SpineJack, or posterior instrumentation? If this is the case, the model was in fact not validated for this study at all, since there are too many assumptions which were not covered by this validation process. Indeed, there is a huge experimental dataset in the literature for the validation of these conditions. Therefore, the authors should consider validating their model for the single conditions before publishing random results based on multiple assumptions. Apart from that, range of motion validation alone is not sufficient when investigating facet joint contact forces, intervertebral disc stress, etc.
Results
Lines 172-173: ‘The results of the ROMs of the present FE model in 10 NM pure moment on the three principal planes were similar to those in the previous studies.’ This is solely partially true, for instance regarding axial rotation. In lateral bending, however, there is a large discrepancy between the present FE model and for example the cadaver model of Raghu et al. (2016), exhibiting only about one third of the experimental range of motion. Apart from that, the presented literature is not referenced.
Lines 173: ‘The results show that the proposed model is reliable.’ This is solely partially true for the range of motion in the intact state, but not for the other conditions as well as the facet joint contact forces, intervertebral disc stress, etc.
Discussion
Lines 227-228: ‘… the cardiopulmonary function is also decreased following the decrease in activity.’ This is indeed an interesting statement. Is there any study confirming this?
Author Response
Very thanks for the questions. We replay all the comments and questions in the attached file.

Reviewer 2 Report
Thank you for the opportunity to review this interesting finite element biomechanical spine study. This study was a biomechanical study to assess how spine mechanics change in the setting of an L3 incomplete burst fracture treated with either complete or incomplete vertebral body height restoration. The fractures were treated either with reduction and cement alone or with spine-jack and cement. Additionally, an analysis for both groups additionally stabilized with a one level dorsal instrumentation was performed. Main findings showed that peak stress at the L2/3 disk as well as forces on the L2/3 facet was significantly higher with incomplete reduction of the fracture. In the setting of dorsal instrumentation, these forces were expectedly decreased at the L2/3 level and marginally increased below the instrumentation.
Overall, this is a well written study with a good study design to address an important biomechanical question of how loss of body height in vertebral compression fractures affects mechanical stresses on the spine. It attempts to help answer how best to treat osteoporotic incomplete burst fractures of the L-spine, a common injury associated with varying degrees of spinal instability. These study results contribute to the spinal surgery knowledge base. Below are my suggestions for improving the manuscript and clarify some issues.
Suggestions for improvement:
One important correction is that the fracture model used in this study is that of an incomplete burst fracture, not a complete burst fracture as suggested in the title and text. Please change accordingly and please also explain why an incomplete burst fracture model was used. At our institution we like using spine-jack with or without dorsal instrumentation in these incomplete burst fractures due to superior fracture fixation without the invasiveness of 360 degree fusion, which would be over treatment in this usually stable fracture pattern.
Also, the discussion needs to be rewritten to focus on the biomechanical aspects of loss of vertebral height. This should also be addressed in the introduction: it is stated that loss of vertebral height is associated with changes in biomechanical properties of the spine, and that this is incompletely understood. However, the research that has been done needs to be added to the introduction and the primary focus of the discussion (the second and fifth paragraphs in the discussion, while well summarized, are not relevant to this biomechanical study and should be significantly shortened or removed).
The conclusion of the study that full vertebral body height restoration should be pursued is stated. However, it would be interesting to discuss that if full body restoration cannot be achieved with spinejack or balloon kyphoplasty alone (knowing the implications on facet joint forces and disk stress), what benefit of adding dorsal instrumentation could have on overall biomechanics. This could be of clinical importance during surgery in helping make a decision for or against dorsal instrumentation after vertebral body reduction and kyphoplasty.
Several limitations to the study:
One major limitation is that a spinal model of a young healthy adult was used. In the setting of an osteoporotic, geriatric spinal fracture with existing degenerative changes, the biomechanical effect of an additional kyphotic deformity in the setting of preexisting deformity, as is likely in this patient population was not assessed in this study. This would be a good direction for future FE biomechanical studies.
Further, FE is obviously a simplified and idealized mathematical simulation, the limitation of applying these results to clinical reality must explicitly be stated.
Finally, please comment on what biomechanical properties were given to the model of the spine-jack implant (Youngs-Modulus, etc) in this simulation. While it is mentioned in the discussion that differences in biomechanical properties were “very minor”, how did the authors come to this conclusion?
Author Response
. .

Round 2
Reviewer 1 Report
The authors did a good job to address most of the comments and questions raised by this reviewer. However, there is still a large lack of self-criticism regarding the used methods, which do not reflect quasi-realistic conditions at all. At least, this reviewer expected that the most obvious limitations were listed and described in detail instead of summarizing them shortly and superficially, which is unfair to the reader in the end. Specifically, the authors should discuss in detail, (1) how the large discrepancies in the biomechanical parameters of previous studies and their model can be explained and (2) what is the difference between real-life kyphoplasty and the simulated one in their model and what are the consequences for the results (see also previous comments of this reviewer). Moreover, a statement if a sensitivity analysis and a verification of the model (not a validation!) were performed, is still missing in the manuscript.
Author Response
Very thanks for the comments. We have revised the manuscript according to the options of reviewer 1. The responses and modified figures are described in the attached documentation.

Round 3
Reviewer 1 Report
Thank you for this interesting work.